Adaptive machine learning approaches utilizing soft decision-making via intuitionistic fuzzy parameterized intuitionistic fuzzy soft matrices

http://orcid.org/0000-0002-0958-5872 Memiş Samet 1 samettmemis@gmail.com
Şola Erduran Ferhan 2
Aydoğan Hivda 3
1 Department of Marine Engineering, Faculty of Maritime, Bandırma Onyedi Eylül University , Balıkesir , Türkiye
2 Department of Mathematics, Faculty of Science, Gazi University , Ankara , Türkiye
3 Department of Mathematics, Graduate School of Natural and Applied Sciences, Gazi University , Ankara , Türkiye
Pamucar Dragan
Electronic publication date: 2025 Feb 28
Publication date: 2025
Volume: 11
Electronic Location ID: e2703
Received 2024 Oct 7; Accepted 2025 Jan 23
Copyright: © 2025 Memiş et al.
Copyright year: 2025
Copyright holder: Memiş et al.
License: This is an open access article distributed under the terms of the Creative Commons Attribution License, which permits unrestricted use, distribution, reproduction and adaptation in any medium and for any purpose provided that it is properly attributed. For attribution, the original author(s), title, publication source (PeerJ Computer Science) and either DOI or URL of the article must be cited.
License URL: https://creativecommons.org/licenses/by/4.0/

Keywords: Intuitionistic fuzzy sets, Soft sets, ifpifs-matrices, Soft decision-making, Machine learning

Funding: The authors received no funding for this work.

==============================
The exponential data growth generated by technological advancements presents significant challenges in analysis and decision-making, necessitating innovative and robust methodologies. Machine learning has emerged as a transformative tool to address these challenges, especially in scenarios requiring precision and adaptability. This study introduces two novel adaptive machine learning approaches, i.e., AIFPIFSC1 and AIFPIFSC2. These methods leverage the modeling ability of intuitionistic fuzzy parameterized intuitionistic fuzzy soft matrices (ifpifs-matrices). This state-of-the-art framework enhances the classification task in machine learning by employing soft decision-making through ifpifs-matrices. The proposed approaches are rigorously evaluated against leading fuzzy/soft-based classifiers using 15 widely recognized University of California, Irvine datasets, including accuracy and robustness, across six performance metrics. Statistical analyses conducted using Friedman and Nemenyi tests further substantiate the reliability and superiority of the proposed approaches. The results consistently demonstrate that these approaches outperform their counterparts, highlighting their potential for solving complex classification problems. This study contributes to the field by offering adaptable and effective solutions for modern data analysis challenges, paving the way for future advancements in machine learning and decision-making systems.

Introduction

The amount of data made possible by technological advancement is constantly growing. This increasing amount of data may be analyzed and interpreted using machine learning, a technical improvement. Numerous industries frequently use this technology, including defense, finance, psychology, medicine, meteorology, astronomy, and space sciences. Such fields are encountered with many uncertainties. Several mathematical tools, such as fuzzy sets (Zadeh, 1965) and intuitionistic fuzzy sets (Atanassov, 1986), have been propounded to model these uncertainties. Furthermore, modeling such uncertainties is a crucial process to enhance the performance of machine learning algorithms. To this end, many classical machine learning algorithms, such as k-nearest neighbor (kNN) (Fix & Hodges, 1951; Keller, Gray & Givens, 1985), have been successfully modified as fuzzy kNN (Keller, Gray & Givens, 1985) using fuzzy sets.

Although fuzzy sets provide a mathematical framework for dealing with uncertainties where classical sets are inadequate, soft sets (Molodtsov, 1999) further this approach by offering additional flexibility in modeling uncertainties. In the last two decades, these concepts have evolved into various hybrid forms such as fuzzy soft sets (Maji, Biswas & Roy, 2001a), fuzzy parameterized soft sets (Çağman, Çıtak & Enginoğlu, 2011), and fuzzy parameterized fuzzy soft sets (fpfs-sets) (Çağman, Çıtak & Enginoğlu, 2010), which have manifested their utility in modeling scenarios where parameters or objects possess fuzzy values.

Developing fuzzy parameterized fuzzy soft matrices (fpfs-matrices) (Enginoğlu & Çağman, 2020) ensures a significant advancement in the field, particularly in computerizing decision-making scenarios involving a large amount of data. It has been applied to performance-based value assignment problems in image denoising by employing generalized fuzzy soft max-min decision-making method (Enginoğlu, Memiş & Çağman, 2019), and operability-configured soft decision-making methods (Enginoğlu et al., 2021; Enginoğlu & Öngel, 2020) and classification problems in machine learning (Memiş & Enginoğlu, 2019). Moreover, classification algorithms based on soft decision-making methods constructed by fpfs-matrices have been suggested, namely FPFS-CMC (Memiş, Enginoğlu & Erkan, 2022a) and FPFS-AC (Memiş, Enginoğlu & Erkan, 2022c). However, fpfs-matrices lack modeling ability for intuitionistic fuzzy uncertainties. Therefore, the concepts of intuitionistic fuzzy soft sets (ifs-sets) (Maji, Biswas & Roy, 2001b), intuitionistic fuzzy parameterized soft sets (ifps-sets) (Deli & Çağman, 2015), and intuitionistic fuzzy parameterized fuzzy soft sets (ifpfs-sets) (El-Yagubi & Salleh, 2013) have been proposed. In addition, intuitionistic fuzzy parameterized intuitionistic fuzzy soft sets (ifpifs-sets) (Karaaslan, 2016) and matrices (ifpifs-matrices) (Enginoğlu & Arslan, 2020) have enabled the modeling of problems with both parameters and objects containing intuitionistic fuzzy uncertainties. Besides, various mathematical tools, such as bipolar soft sets (Mahmood, 2020), intuitionistic fuzzy mean operators (Hussain et al., 2023), and picture fuzzy soft matrices (Memiş, 2023b), have been proposed to deal with uncertainty and are still being studied. However, when the related literature is investigated, it is noteworthy that there are almost no applications, especially in real-world problems/data (Bustince Sola et al., 2016; Karakoç, Memiş & Sennaroglu, 2024). Although several applications have recently been conducted to the classification problem in machine learning, which is a real problem modeled by fpfs-matrices, only a few studies employ ifpifs-matrices in machine learning (Memiş et al., 2021). Therefore, applying ifpifs-matrices, a pioneer mathematical tool to model uncertainty, to machine learning is a topic worthy of study.

Recently, Memiş et al. (2021) have defined metrics, quasi-metrics, semi-metrics, and pseudo-metrics, as well as similarities including quasi-, semi-, and pseudo-similarities over ifpifs-matrices. Besides, Memiş et al. (2021) has proposed an Intuitionistic Fuzzy Parameterized Intuitionistic Fuzzy Soft Classifier (IFPIFSC), which utilizes six pseudo-similarities of ifpifs-matrices. This classifier has been simulated using 20 datasets from the University of California, Irvine Machine Learning Repository (UCI-MLR) (Kelly, Longjohn & Nottingham, 2024). Its performance has been evaluated using six metrics: accuracy (Acc), precision (Pre), recall (Rec), specificity (Spe), macro F-score (MacF), and micro F-score (MicF). The results have manifested that IFPIFSC has outperformed fuzzy and non-fuzzy-based classifiers in these metrics. Although machine learning algorithms have been suggested via ifpifs-matrices for real problems, machine learning datasets in UCI-MLR and other databases usually consist of real-valued raw data. In these datasets, no serious uncertainties, such as intuitionistic uncertainty, can be modeled with mathematical tools. However, our aim in this study is to propose a new machine learning algorithm that can work with a dataset that contains intuitionistic fuzzy uncertainties by first converting the raw data to fuzzy values and then to intuitionistic fuzzy values. On the other hand, the most significant disadvantage of the two algorithms described above based on ifpifs-matrices, i.e., IFPIFSC and IFPIFS-HC, is that working by fixed λ1 and λ2 values. In this article, instead of fixed λ1 and λ2 values, we aim to develop an equation that allows the algorithm to determine these values based on the dataset and a machine learning algorithm that can work adaptively using this equation. Moreover, the soft decision-making methods constructed by ifpifs-matrices, one of the efficacious decision-making techniques, can be integrated into the machine learning process. As a result, in this study, we focus on proposing a machine learning algorithm based on pseudo-similarities of ifpifs-matrices, adaptive λ1 and λ2 values for intuitionistic fuzzification of the data, and soft decision-making methods constructed by ifpifs-matrices. The significant contributions of the article can be summed up as follows: Improving two adaptive machine learning algorithms employing ifpifs-matrices,

Utilizing two soft decision-making methods constructed by ifpifs-matrices in machine learning.

The fact that this article is one of the pioneer studies combining soft sets, intuitionistic fuzzy sets, and machine learning.

Applying the similarity measures of ifpifs-matrices to classification problems in machine learning, in contrast to many soft set-based studies on hypothetical problems.

A key innovation in our approach is the adaptive modification of lambda values in the classification algorithms, which significantly enhances the adaptability and Acc of the classifiers. By integrating ifpifs-matrices with two classification algorithms and dynamically adjusting lambda values, our method represents a novel contribution to machine learning. This adaptive approach not only addresses the limitations of previous classifiers but also demonstrates superior performance in handling uncertainties and dynamic natures of real-world data, setting a new benchmark for future research in the domain.

The rest of the present study is organized as follows: The second section provides some fundamental definitions, notations, and algorithms to be needed for the following sections. The third section presents the basic definitions for the two proposed algorithms and their algorithmic designs. The fourth section details the utilized datasets and performance metrics. Secondly, it simulates the proposed algorithms with the well-known and state-of-the-art fuzzy and soft sets/matrices-based classification algorithms. Finally, it statistically analyzes the simulation results using the Friedman and Nemenyi tests. The final section discusses the proposed approaches, their performance results, and the need for further research.

Preliminaries

This section first introduces the notion of ifpifs-matrices (Enginoğlu & Arslan, 2020) and several of its fundamental characteristics. During this research, let E and U represent a parameter set and an alternative (object) set, respectively.

Definition 1 (Atanassov, 1986) Let μ and ν represent two functions from E to [0,1] such that μ(x)+ν(x)≤1, for all x∈E. The set {(x,μ(x),ν(x)):x∈E} is referred to as an intuitionistic fuzzy set (if-set) over E.

Here, for all x∈E, μ(x) and ν(x) are called the membership and non-membership degrees, respectively. In addition, the indeterminacy degree of x is defined by π(x)=1−(μ(x)+ν(x)).

Across the study, the set of all the if-sets over E is denoted by IF(E) and f∈IF(E). Briefly, the notation xν(x)μ(x) can be employed instead of (x,μ(x),ν(x)). Thus, an if-set over E can be denoted by f={xν(x)μ(x):x∈E}.

Definition 2 (Karaaslan, 2016) Let f∈IF(E) and α be a function from f to IF(U). Then, the set {(xν(x)μ(x),α(ν(x)μ(x)x)):x∈E} is called an ifpifs-set parametrized via E over U (or over U for brevity) and denoted by α.

In the present study, the set of all the ifpifs-sets over U is denoted by IFPIFSE(U).

Definition 3 (Enginoğlu & Arslan, 2020) Let α∈IFPIFSE(U). Then, [aij] is called ifpifs-matrix of α and is defined by

[aij]:=[a01a02a03⋯a0na11a02a03⋯a1n⋮⋮⋮⋱⋮am1am2am3⋯amn]

such that i∈{0,1,2,…} and j∈{1,2,…}, aij:={ν(x)μ(x),i=0α(ν(x)μ(x)xj)(ui),i≠0 or briefly aij:=νijμij. Here, if |E|=n and |U|=m−1, then [aij] is an m×n ifpifs-matrix.

Hereinafter, the set of all the ifpifs-matrices over U is denoted by IFPIFSE[U].

Secondly, we provide iCCE10 (Arslan et al., 2021) and isMBR01 (Arslan et al., 2021) in Algorithms 1 and 2, respectively, by considering the notations used across this study.

Algorithm 1 Pseudocode of iCCE10.

Input: ifpifs-matrix [aij]m×n	
Output: Score matrix [si1](m−1)×1, Optimum alternatives [opi1]	
1:    [μ]←[0](m−1)×1	
2:   [ν]←[0](m−1)×1	
3:  for i from 2 to m do	
4:    for j from 1 to n do	
5:      μi1←μi1+a01jaij1	
6:      νi1←νi1+a01jaij2	
7:    end for	
8:      si1←1nμi1	
9:      si2←1nνi1	
10:  end for	
11:   [sv]←[si1−si2](m−1)×1	
12:   [av]←[si1+si2](m−1)×1	
13:  for i from 1 to m−1 do	
14:   if maxk∈Im−1⁡{svk1}=0 AND mink∈Im−1⁡{svk1}=0 then	
15:     dmi1←1	
16:   else	
17:      dmi1←svi1+|mink∈Im−1⁡{svk1}|maxk∈Im−1⁡{svk1}+|mink∈Im−1⁡{svk1}|	
18:   end if	
19:  end for	
20:  [op]←argmaxk∈Im−1⁡{svk1}	

Algorithm 2 Pseudocode of is MBR01.

Input: ifpifs-matrix [aij]m×n	
Output: Score matrix [si1](m−1)×1, Optimum alternatives [opi1]	
1:   [b]←[0](m−1)×1 and [c]←[0](m−1)×1	
2:   for i from 1 to m−1 do	
3:    for k from 1 to m−1 do	
4:      for j from 1 to n do	
5:       bi1←bi1+a1j1sgn(aij1−akj1)	
6:       ci1←ci1+a1j2sgn(aij2−akj2)	
7:      end for	
8:    end for	
9:   end for	
10:  [s]←[0](m−1)×1×2	
11: for i from 1 to m−1 do	
12:    if maxk∈Im−1⁡{bk1}+|maxk∈Im−1⁡{ck1}|+|mink∈Im−1⁡{bk1}|≠0 AND mink∈Im−1⁡{bk1}≠0 then	
13:     si11←bi1+|mink∈Im−1⁡{bk1}|maxk∈Im−1⁡{bk1}+|maxk∈Im−1⁡{ck1}|+|mink∈Im−1⁡{bk1}|	
14:     si12←1−bi1+|ci1|+|mink∈Im−1⁡{bk1}|maxk∈Im−1⁡{bk1}+|maxk∈Im−1⁡{ck1}|+|mink∈Im−1⁡{bk1}|	
15:    else	
16:      si11←1	
17:      si12←0	
18:    end if	
19: end for	
20:  [sv]←[0](m−1)×1	
21: for i from 1 to m−1 do	
22:    svi1←si11−si12	
23: end for	
24:  [av]←[0](m−1)×1	
25: for i from 1 to m−1 do	
26:    avi1←si11+si12	
27: end for	
28:  [op]←argmaxk∈Im−1⁡{svk1}	

Proposed adaptive machine learning approaches

This section overviews the fundamental mathematical notations essential for the proposed classifier based on ifpifs-matrices. In this article, we represent the data with a matrix D=[dij]m×(n+1), where m represents the number of samples, n is the number of parameters, and the last column of D contains the labels for the data. The training data matrix, denoted as (Dtrain)m1×n, along with the corresponding class matrix Cm1×1, is used to generate a testing matrix (Dtest)m2×n derived from the original data matrix D, where m1+m2=m. Additionally, we employ the matrix Uk×1 to stand for the unique class labels extracted from Cm1×1. Notably, Di−train and Di−test refer to the i-th rows of Dtrain and Dtest, respectively. Similarly, Dtrain−j and Dtest−j denote the j-th columns of Dtrain and Dtest. Furthermore, Tm2×1′ represents the predicted class labels for the testing samples.

Definition 4 Let x,y∈Rn. Then, P:Rn×Rn→[−1,1] is a function defined by

P(x,y):=n∑j=1nxjyj−(∑j=1nxj)(∑j=1nyj)[n∑j=1nxj2−(∑j=1nxj)2][n∑j=1nyj2−(∑j=1nyj)2].

is denoted as the Pearson correlation coefficient for the variables x and y.

Definition 5 Let x∈Rn and j∈In. The normalizing vector of x is defined as x^∈Rn, where

x^j:={xj−mink∈In⁡{xk}maxk∈In⁡{xk}−mink∈In⁡{xk},maxk∈In⁡{xk}≠mink∈In⁡{xk}1,maxk∈In⁡{xk}=mink∈In⁡{xk}.

Definition 6 Let D=[dij]m×(n+1) be a data matrix, i∈Im, and j∈In. A column normalized matrix of D is defined by D~=[d~ij]m×n, where

d~ij:={dij−mink∈Im⁡{dkj}maxk∈Im⁡{dkj}−mink∈Im⁡{dkj},maxk∈Im⁡{dkj}≠mink∈Im⁡{dkj}1,maxk∈Im⁡{dkj}=mink∈Im⁡{dkj}.

Definition 7 Let (Dtrain)m1×n be a training matrix obtained from D=[dij]m×(n+1), i∈Im1, and j∈In. A column normalized matrix of Dtrain is defined by D~train=[d~ij−train]m1×n, where

d~ij−train:={dij−train−mink∈Im⁡{dkj}maxk∈Im⁡{dkj}−mink∈Im⁡{dkj},maxk∈Im⁡{dkj}≠mink∈Im⁡{dkj}1,maxk∈Im⁡{dkj}=mink∈Im⁡{dkj}.

Definition 8 Let (Dtest)m2×n be a testing matrix obtained from D=[dij]m×(n+1), i∈Im2, and j∈In. A column normalized matrix of Dtest is defined by D~test=[d~ij−test]m1×n, where

d~ij−test:={dij−test−mink∈Im⁡{dkj}maxk∈Im⁡{dkj}−mink∈Im⁡{dkj},maxk∈Im⁡{dkj}≠mink∈Im⁡{dkj}1,maxk∈Im⁡{dkj}=mink∈Im⁡{dkj}.

Definition 9 (Memiş et al., 2021) Let Dtrain=[dij−train]m1×n and Cm1×n be a training matrix and its class matrix obtained from a data matrix D=[dij]m×(n+1), respectively. Then, the matrix ifwDtrainλP=[μ1jλPν1jλP]1×n is called intuitionistic fuzzification weight (ifw) matrix based on Pearson correlation coefficient of Dtrain and defined by

μ1jλP:=1−(1−|P(Dtrain−j,C)|)λ

and

ν1jλP:=(1−|P(Dtrain−j,C)|)λ(λ+1)

such that j∈In and λ∈[0,∞).

Definition 10 (Memiş et al., 2021) Let D~train=[d~ij−train]m1×n be a column normalized matrix of a matrix (Dtrain)m1×n. Then, the matrix D~~trainλ=[d~~train−ijλ]=[μij−trainD~~λνij−trainD~~λ]m1×n is called intuitionistic fuzzification of D~train and defined by

μij−trainD~~λ:=1−(1−d~ij−train)λ

and

νij−trainD~~λ:=(1−d~ij−train)λ(λ+1)

such that i∈Im1, j∈In, and λ∈[0,∞).

Definition 11 (Memiş et al., 2021) Let D~test=[d~ij−test]m2×n be a column normalized matrix of a matrix (Dtest)m2×n. Then, the matrix D~~testλ=[d~~test−ijλ]=[μij−testD~~λνij−testD~~λ]m2×n is called intuitionistic fuzzification of D~test and defined by

μij−testD~~λ:=1−(1−d~ij−test)λ

and

νij−testD~~λ:=(1−d~ij−test)λ(λ+1)

such that i∈Im2, j∈In, and λ∈[0,∞).

Definition 12 (Memiş et al., 2021) Let (D~train)m1×n be a column normalized matrix of a matrix (Dtrain)m1×n and D~~trainλ=[d~~train−ijλ]=[μij−trainD~~λνij−trainD~~λ]m1×n be the intuitionistic fuzzification of D~train. Then, the ifpifs-matrix [bijD~~k−trainλ]2×n is called the training ifpifs-matrix obtained by k-th row of D~~trainλ and ifwDtrainλP and defined by

b0jD~~k−trainλ:=ν1jλPμ1jλPandb1jD~~k−trainλ:=νkj−trainD~~λμkj−trainD~~λ

such that k∈Im1 and j∈In.

Definition 13 (Memiş et al., 2021) Let (D~test)m2×n be a column normalized matrix of a matrix (Dtest)m2×n and D~~testλ=[d~~test−ijλ]=[μij−testD~~λνij−testD~~λ]m2×n be the intuitionistic fuzzification of D~test. Then, the ifpifs-matrix [aijD~~k−testλ]2×n is called the testing ifpifs-matrix obtained by k-th row of D~~testλ and ifwDtrainλP and defined by

a0jD~~k−testλ:=ν1jλPμ1jλPanda1jD~~k−testλ:=νkj−testD~~λμkj−testD~~λ

such that k∈Im1 and j∈In.

Secondly, it presents the concept of pseudo-similarities over IFPIFSE[U] and seven pseudo-similarities over IFPIFSE[U].

Definition 14 (Memiş et al., 2021) Let s~:IFPIFSE[U]×IFPIFSE[U]→R be a mapping. Then, s~ is a pseudo-similarity over IFPIFSE[U] if and only if s~ satisfies the following properties for all [aij], [bij], ∈IFPIFSE[U]: (i) s~([aij],[aij])=1,

(ii) s~([aij],[bij])=s~([bij],[aij]),

(iii) 0≤s~([aij],[bij])≤1.

Thirdly, this part provides the Minkowski, Hamming, Euclidean, Hamming-Hausdorff, Chebyshev, Jaccard, and Cosine pseudo-similarities over IFPIFSE[U] using the normalized pseudo-metrics of ifpifs-matrices (For more details see (Memiş et al., 2023)).

Proposition 15 (Memiş et al., 2023) Let p∈Z+. Then, the mapping s~Mp:IFPIFSE[U]×IFPIFSE[U]→R defined by

s~Mp([aij],[bij]):=1−(12∑i=1m−1∑j=1n(|μ0jaμija−μ0jbμijb|p+|ν0jaνija−ν0jbνijb|p+|π0jaπija−π0jbπijb|p)1p

is a pseudo-similarity and referred to as Minkowski pseudo-similarity.

In this case, s~M1 and s~M2 are represented by s~H and s~E, respectively, and are referred to as Hamming pseudo-similarity (Memiş et al., 2021) and Euclidean pseudo-similarity.

Proposition 16 (Memiş et al., 2023) Let p∈Z+. Then, the mapping s~Hs:IFPIFSE[U]×IFPIFSE[U]→R defined by

s~Hs([aij],[bij]):=1−12(m−1)∑i=1m−1maxj∈In{|μ0jaμija−μ0jbμijb||ν0jaνija−ν0jbνijb|+|π0jaπija−π0jbπijb|}

is a pseudo-similarity and referred to as Hamming-Hausdorff pseudo-similarity.

Proposition 17 (Memiş et al., 2023) Let p∈Z+. Then, the mapping s~Ch:IFPIFSE[U]×IFPIFSE[U]→R defined by

s~Ch([aij],[bij]):=1−maxi∈Im−1⁡{maxj∈In⁡{|μ0jaμija−μ0jbμijb|+|ν0jaνija−ν0jbνijb|+|π0jaπija−π0jbπijb|}}

is a pseudo-similarity and is referred to as Chebyshev pseudo-similarity.

Proposition 18 (Memiş et al., 2023) The mapping s~J:IFPIFSE[U]×IFPIFSE[U]→R defined by

s~J([aij],[bij]):=1m−1∑i=1m−1ϵ+xiϵ+yi+zi−xi

such that

xi=∑j=1nμ0jaμijaμ0jbμijb+ν0jaνijaν0jbνijb+π0jaπijaπ0jbπijb

yi=∑j=1n(μ0jaμija)2+(ν0jaνija)2+(π0jaπija)2

and

zi=∑j=1n(μ0jbμijb)2+(ν0jbνijb)2+(π0jbπijb)2

is known as the Jaccard pseudo-similarity, and it is a pseudo-similarity. Here, 0≪ϵ< 1, for example, ϵ=0.000001.

Proposition 19 (Memiş et al., 2023) The mapping s~C:IFPIFSE[U]×IFPIFSE[U]→R defined by

s~C([aij],[bij]):=1m−1∑i=1m−1ϵ+xiϵ+yizi

such that

xi=∑j=1nμ0jaμijaμ0jbμijb+ν0jaνijaν0jbνijb+π0jaπijaπ0jbπijb

yi=∑j=1n(μ0jaμija)2+(ν0jaνija)2+(π0jaπija)2

and

zi=∑j=1n(μ0jbμijb)2+(ν0jbνijb)2+(π0jbπijb)2

is known as the Cosine pseudo-similarity, and it is a pseudo-similarity. Here, 0≪ϵ< 1, for example, ϵ=0.000001.

Fourthly, it propounds the classification algorithms AIFPIFSC1 and AIFPIFSC2 and provides the pseudocodes of normalize and intuitionistic normalize functions in Algorithms 3 and 4 to be needed for the proposed algorithms’ pseudocodes (Algorithms 5 and 6).

Algorithm 3 Normalize function.

Input: am×n	
Output: a~m×n	
1:  function normalize(a)	
2:      [m,n]←size(a)	
3:     if max(a)≠min(a) then	
4:        a~←(a−min(a))/(max(a)−min(a))	
5:     else	
6:        a~← ones(m, n)	
7:     end if	
8:     return a~	
9:  end function	

Algorithm 4 Intuitionistic normalize function.

Input: am×n, λ	
Output: a~~m×n×2	
1: function inormalize (a,λ)	
2:    [m,n]←size(a)	
3:    for i=1 to m do	
4:     for j=1 to n do	
5:       a~~(i,j,1)←1−(1−a(i,j))λ	
6:       a~~(i,j,2)←(1−a(i,j))λ(λ+1)	
7:     end for	
8:    end for	
9:    return a~~	
10: end function	

Algorithm 5 AIFPIFSC1’s pseudocode algorithm.

Input: (Dtrain)m1×n, Cm1×1, and (Dtest)m2×n	
Output: Tm2×1′	
1: function AIFPIFSC1(train, C, test)	
2:    λ1←round(n)	
3:     λ2←round(ln⁡(n)/ln⁡(length(unique(C))))	
4:    Compute ifwDtrainλ1P using Dtrain,C and λ1	
5:    Compute feature fuzzification of Dtrain and Dtest, namely D~train and D~test	
6:    Compute D~~trainλ1 and D~~testλ1	
7:    for i from 1 to tm do	
8:      Compute test ifpifs-matrix [aij] using if wDtrainλ1P and D~~k−testλ1	
9:      for j from 1 to em do	
10:       Compute train ifpifs-matrix [bij] using if wDtrainλ1P and D~~k−trainλ1	
11:       cm(j,1)←s~H([aij],[bij])	
12:       cm(j,2)←s~Hs([aij],[bij])	
13:       cm(j,3)←s~Ch([aij],[bij])	
14:       cm(j,4)←s~E([aij],[bij])	
15:       cm(j,5)←s~M3([aij],[bij])	
16:         cm(j,6)←s~J([aij],[bij])	
17:        cm(j,7)←s~C([aij],[bij])	
18:     end for	
19:     for j from 1 to size(cm,2) do	
20:       sd(1,j)←std(cm(:,j))	
21:     end for	
22:     wm2(1,:)←1−NORMALIZE(sd/4)	
23:     dm←[wm2;cm]	
24:     idm←INORMALIZE(dm,λ2)	
25:     [,,,op]←isMBR01(idm)	
26:     ti1′←C(op(1),1)	
27:   end for	
28:  end function	

Algorithm 6 AIFPIFSC2’ pseudocode.

Input: (Dtrain)m1×n, Cm1×1, and (Dtest)m2×n	
Output: Tm2×1′	
1: function AIFPIFSC2(train, C, test)	
2:     λ1←round(n)	
3:     λ2←round(ln⁡(n)/ln⁡(length(unique(C))))	
4:    Compute ifwDtrainλ1P using Dtrain,C and λ1	
5:    Compute feature fuzzification of Dtrain and Dtest, namely D~train and D~test	
6:    Compute D~~trainλ1 and D~~testλ1	
7:    for i from 1 to tm do	
8:     Compute test ifpifs-matrix [aij] using ifwDtrainλ1P and D~~k−testλ1	
9:     for j from 1 to em do	
10:       Compute train ifpifs-matrix [bij] using ifwDtrainλ1P and D~~k−trainλ1	
11:       cm(j,1)←s~H([aij],[bij])	
12:       cm(j,2)←s~Hs([aij],[bij])	
13:       cm(j,3)←s~Ch([aij],[bij])	
14:       cm(j,4)←s~E([aij],[bij])	
15:       cm(j,5)←s~M3([aij],[bij])	
16:       cm(j,6)←s~J([aij],[bij])	
17:       cm(j,7)←s~C([aij],[bij])	
18:     end for	
19:     for j from 1 to size(cm,2) do	
20:       sd(1,j)←std(cm(:,j))	
20:     end for	
21:      wm2(1,:)←1−NORMALIZE(sd/4)	
22:      dm←[wm2;cm]	
23:      idm←INORMALIZE(dm,λ2)	
24:      [,,,op]←iCCE10(idm)	
25:      ti1′←C(op(1),1)	
26:   end for	
27: end function	

In the algorithm AIFPIFSC1, parameters λ1 and λ2 are determined based on the dataset’s characteristics. Then, ifw is computed by measuring the Pearson correlation coefficient between each feature and the class labels. These weights are utilized to construct two ifpifs-matrices: one for training data and the other for testing data. Feature fuzzification of the training data features is obtained using the ifw. For each testing sample, a comparison matrix is constructed by calculating pseudo-similarities between the testing ifpifs-matrix and training ifpifs-matrix. This comparison matrix serves as the basis for parameter weights’ computation. Parameter weights are determined by calculating the standard deviation of each column in the comparison matrix. A comparison ifpifs-matrix is then generated by combining these parameter weights with the comparison matrix. Next, we apply the algorithm isMBR01 to this comparison ifpifs-matrix to identify the optimum training sample. The class label of this optimum training sample is assigned to the corresponding testing sample. These steps are repeated for all testing samples to obtain the predicted class labels.

The similar steps in Algorithm 6 are repeated for the algorithm AIFPIFSC2. In the last part, we apply the algorithm iCCE10 to the aforesaid comparison ifpifs-matrix to determine the optimum training sample.

Finally, an illustrative example is presented below to enhance the clarity of the AIFPIFSC1 algorithm concerning its pseudocode. In this study, we use k-fold cross-validation to avoid the possibility of the algorithm’s results being overly optimistic in a single run.

Illustrative Example:

A data matrix D=[dij]10×5 from the “Iris” dataset (Fisher, 1936) is provided below for implementing AIFPIFSC1. The matrix contains 10 samples divided into three classes ( l=3), with the class labels in the last column. Class one consists of three samples, class two has three samples, and class three includes four. In the first iteration of five-fold cross-validation, (Dtrain)8×4, C8×1, (Dtest)2×4, and T2×1 are obtained as follows:

D=[5.13.51.40.214.93.01.40.214.73.21.30.217.03.24.71.426.43.24.51.526.93.14.91.526.33.36.02.535.82.75.11.937.13.05.92.136.32.95.61.83]

Dtrain=[5.13.51.40.24.93.01.40.24.73.21.30.27.03.24.71.46.93.14.91.56.33.36.02.55.82.75.11.97.13.05.92.1]C=[11122333]Dtest=[6.43.24.51.56.32.95.61.8]T=[23].

Dtrain, C, and Dtest are inputted to AIFPIFSC1. After the classification task, the ground truth class matrix T is utilized to calculate the Acc, Pre, Rec, Spe, MacF, and MicF rates of AIFPIFSC1.

Secondly, λ1 and λ2 are calculated as 2 and 1, respectively.

Thirdly, the ifw matrix is calculated via Dtrain and C concerning the Pearson correlation coefficient. The main goal of obtaining the ifw matrix is to construct the train and test ifpifs-matrices in the following phases.

[ifwij]=[0.91100.00070.70260.02630.997500.99940].

Thirdly, feature fuzzifications of Dtrain and Dtest are computed as follows:

D~train=[0.166710.021300.08330.37500.0213000.6250000.95830.62500.72340.52170.91670.50000.76600.56520.66670.7500110.458300.80850.739110.37500.97870.8261]D~test=[0.70830.62500.68090.56520.66670.25000.91490.6957].

Fourthly, intuitionistic feature fuzzifications of D~train and D~test are computed as follows:

D~~train=[0.16670.69441.00000.00000.02130.95790.00001.00000.08330.84030.37500.39060.02130.95790.00001.00000.00001.00000.62500.14060.00001.00000.00000.00000.95830.00170.62500.14060.72340.07650.52170.22870.91670.00000.50000.25000.76600.05480.56520.18900.66670.11110.75000.06251.00000.00001.00000.00000.45830.29340.00001.00000.80850.03670.73910.06811.00000.84030.37500.39060.97870.00050.82610.0302]D~~test=[0.70830.08510.62500.14060.68091.01900.56520.18900.66670.11110.25000.56250.91490.00720.69570.0926].

In the next steps, D~~train and D~~test are required to constructing the train and test ifpifs-matrices.

For i=1, the following steps are performed:

Fifthly, for all j, test ifpifs-matrix [aij] and jth train ifpifs-matrix [bij] are constructed by utilizing D~i−test and D~j−train. For instance ( j=1), for D~1−test and D~1−train, test ifpifs-matrix [aij] and 1-train ifpifs-matrix [bij] are constructed as follows:

[aij]=[0.91100.00070.70260.02630.997500.999400.70830.08510.62500.14060.68090.10190.56520.1890][bij]=[0.91100.00070.70260.02630.997500.999400.16670.6944100.02130.95790        1].

The first rows of [aij] and [bij] are the intuitionistic feature weights obtained in the third step. Second rows of [aij] and [bij] are the first row (first sample) of D~~test and first row (first sample) of D~~train, respectively.

Sixtly, cm11, cm12, cm13, cm14, cm15, cm16, and cm17 of CM are computed by employing the pseudo-similarities of the aforesaid [aij] and [bij] as follows:

cm11=s~H([aij],[bij])=0.7432cm12=s~Hs([aij],[bij])=0.6708cm13=s~Ch([aij],[bij])=0.3416cm14=s~E([aij],[bij])=0.6345cm15=s~M3([aij],[bij])=0.9742cm16=s~J([aij],[bij])=0.2826cm17=s~C([aij],[bij])=0.4960.

Then, CM is calculated as follows:

CM=[0.74320.67080.34160.63450.97420.28260.49600.75160.67080.34160.62800.97190.14200.54950.76140.66010.32030.61320.96720.14130.37580.95890.87870.75750.91650.99950.96520.98870.95230.89930.79870.92010.99970.96780.98930.88810.78270.56530.80650.99520.86190.97360.86770.73740.47470.80770.99560.80680.89310.87190.85120.70240.82040.99720.87680.9766].

Seventhly, the standard deviation matrix [sd1j] and second weight matrix [wm1j] are computed as follows:

[sd1j]=[0.08740.09860.19730.12670.01390.37230.2601]

and

[wm1j]=[0.79470.76340.48820.6851100.3130].

Eighthly, the decision matrix DM is obtained by employing CM and [wm1j] as follows:

DM=[0.79470.76340.48820.6851100.31300.74320.67080.34160.63450.97420.28260.49600.75160.67080.34160.62800.97190.14200.54950.76140.66010.32030.61320.96720.14130.37580.95890.87870.75750.91650.99950.96520.98870.95230.89930.79870.92010.99970.96780.98930.88810.78270.56530.80650.99520.86190.97360.86770.73740.47470.80770.99560.80680.8931].

Ninthly, the intuitionistic decision matrix iDM=inormalize(DM,λ2) is computed as follows:

iDM=[0.79470.04210.76340.05600.48820.26190.68510.09921.00000.00000.00001.00000.31300.47190.74320.06590.67080.10840.34160.43350.63450.13360.97420.00070.28260.51460.49600.25410.75160.06170.67080.10840.34160.43350.62800.13840.97190.00080.14200.73610.54950.20290.76140.05690.66010.11550.32030.46200.61320.14960.96720.00110.14130.73740.37580.38960.95890.00170.87870.01470.75750.05880.91650.00700.99950.00120.96520.00100.98870.00010.95230.01250.89930.04720.79870.18890.92010.03740.99970.00000.96780.01910.98930.00070.88810.01630.78270.04180.56530.27590.80650.04300.99520.00000.86190.03730.97360.01140.86770.01640.73740.02210.47470.08860.80770.03230.99560.00000.80680.01520.89310.0005].

Tenthly, the soft decision-making algorithm isMBR01 based on ifpifs-matrices is applied to iDM and determines the optimum train sample.

Finally, the label of the optimum train sample is assigned to ti1′. Because the optimum train sample is the fifth, its label 2 is assigned to t11′.

If t21′ is calculated as t11′ is, then the predicted class matrix t2×1′ is attained as T′=[23]. According to these results, the Acc rate of the proposed algorithm is 100% for the considered example.

Simulation and performance comparison

The current section provides a comprehensive overview of the 15 datasets within the UCI-MLR for classification tasks. Six performance metrics that can be used to compare performances are then presented. Subsequently, a simulation is conducted to illustrate that AIFPIFSC1 and AIFPIFSC2 outperform Fuzzy kNN, FPFS-kNN, FPFS-AC, FPFS-EC, FPFS-CMC, IFPIFSC, and IFPIFS-HC in terms of classification performance. Furthermore, the section conducts statistical analyses on the simulation results using the Friedman test, a non-parametric test, and the Nemenyi test, a post hoc test.

UCI datasets and features

This subsection provides the characteristics of the datasets utilized in the simulation, as outlined in Table 1. The datasets in Table 1 can be accessed from the UCI-MLR (Kelly, Longjohn & Nottingham, 2024).

Table 1 Descriptions of UCI datasets.

No.	Ref.	Name	#Instance	#Attribute	#Class	Balanced/Imbalanced	
1	Nakai (1996)	Ecoli	336	7	8	Imbalanced	
2	Silva & Maral (2013)	Leaf	340	14	36	Imbalanced	
3	Dias, Peres & Bscaro (2009)	Libras	360	90	15	Balanced	
4	Higuera, Gardiner & Cios (2015)	Mice	1,077	72	8	Imbalanced	
5	Lichtinghagen, Klawonn & Hoffmann (2020)	HCV Data	589	12	5	Imbalanced	
6	Barreto & Neto (2005)	Column3C	310	6	3	Imbalanced	
7	Quinlan (1986)	NewThyroid	215	5	3	Imbalanced	
8	Deterding, Niranjan & Robinson (1988)	Vowel	990	13	11	Imbalanced	
9	Fisher (1936)	Iris	150	4	3	Imbalanced	
10	Charytanowicz et al. (2010)	Seeds	210	7	3	Balanced	
11	Bhatt (2017)	Wireless	2,000	7	4	Balanced	
12	Aeberhard & Forina (1992)	Wine	178	13	3	Imbalanced	
13	Cardoso (2013)	WholesaleR	440	6	3	Imbalanced	
14	JP & Jossinet (1996)	Breast Tissue	106	9	6	Imbalanced	
15	Breiman et al. (1984)	Led7Digit	500	7	7	Imbalanced	
Note:

# represents “the number of”.

The main problems related to each dataset in Table 1 are as follows: In “Ecoli” predicting the localization site of Ecoli proteins based on biological attributes. In “Leaf”, different species of plants are classified using leaf shape and texture features. In “Libras, ” accelerometer data recognizes hand movements in Libras (a Brazilian sign language). In “Mice”, gene expression data is analyzed to distinguish between mouse strains. In “HCV Data”, detect Hepatitis C Virus (HCV) infection stages using blood test data. In “Column 3C”, diagnosing spinal disorders by analyzing biomechanical features of vertebrae. In “NewThyroid”, thyroid conditions (e.g., normal or hypothyroid) are classified based on medical attributes. In “Vowel”, spoken vowels are classified using speech acoustics data. In “Iris”, identify iris species based on flower petal and sepal dimensions. In “Seeds”, classify types of wheat seeds based on their geometric properties. In “Wireless”, predict indoor locations based on Wi-Fi signal strength features. In “Wine”, distinguish wine varieties based on their chemical compositions. In “Wholesale”, predict customer segments based on annual spending on various product categories. In “Breast Tissue”, types of breast tissue samples (e.g., healthy or tumor) are classified using bioimpedance data. In “Led7Digit”, human errors in recognizing handwritten digits are simulated on a seven-segment display.

Performance metrics

This subsection outlines the mathematical notations for six performance metrics (Erkan, 2021; Fawcett, 2006; Nguyen et al., 2019), namely Acc, Pre, Rec, Spe, MacF, and MicF, used to compare the mentioned classifiers. Let Dtest={x1,x2,…,xn}, T={t1,t2,…,tn}, T′={t1′,t2′,…,tn′}, and l be n samples for classification, ground truth class sets of the samples, prediction class sets of the samples, and the number of classes for the samples, respectively. Here,

Acc(T,T′):=1l∑i=1lTPi+TNiTPi+TNi+FPi+FNiPre(T,T′):=1l∑i=1lTPiTPi+FPiRec(T,T′):=1l∑i=1lTPiTPi+FNiSpe(T,T′):=1l∑i=1lTNiTNi+FPiMacF(T,T′):=1l∑i=1l2TPi2TPi+FPi+FNiMicF(T,T′):=2∑i=1lTPi2∑i=1lTPi+∑i=1lFPi+∑i=1lFNi

where the numbers for true positive, true negative, false positive, and false negative for the class i are, respectively, TPi, TNi, FPi, and FNi, respectively. Their mathematical notations are as follows:

TPi:=|{xk|i∈Tk∧i∈Tk′,1≤k≤l}|TNi:=|{xt|i∉Tk∧i∉Tk′,1≤k≤l}|FPi:=|{xt|i∉Tk∧i∈Tk′,1≤k≤l}|FNi:=|{xt|i∈Tk∧i∉Tk′,1≤k≤l}|.

Here, the notation |⋅| denotes the cardinality of a set.

Simulation results

This subsection conducts a comparative analysis between AIFPIFSC1 and AIFPIFSC2 and well-established classifiers that rely on fuzzy and soft sets, such as Fuzzy kNN, FPFS-kNN, FPFS-AC, FPFS-EC, FPFS-CMC, IFPIFSC, and IFPIFS-HC. The comparison is performed using MATLAB R2021b (The MathWorks, Natick, NY, USA) on a laptop with an Intel(R) Core(TM) i5-10300H CPU @ 2.50 GHz and 8 GB RAM. The mean performance results of the classifiers are derived from random 10 independent runs based on five-fold cross-validation (Stone, 1974). In each cross-validation iteration, the dataset is randomly divided into five parts, whose four parts are used for training and the remaining part for testing (for further details on k-fold cross-validation, refer to Stone (1974)). Table 2 presents the average Acc, Pre, Rec, Spe, MacF, and MicF results for AIFPIFSC1, AIFPIFSC2, Fuzzy kNN, FPFS-kNN, FPFS-AC, FPFS-EC, FPFS-CMC, IFPIFSC, and IFPIFS-HC across the datasets.

Table 2 Comparative simulation results of the aforesaid classifiers.

Datasets	Classifiers	Acc	Pre	Rec	Spe	MacF	MicF	
Ecoli	Fuzzy kNN	92.1824	53.5997	60.2825	95.6846	66.3808	68.7296	
	FPFS-kNN	94.4340	72.9682	65.3519	96.4741	74.9499	81.1010	
	FPFS-AC	94.2172	72.7755	68.6908	96.3172	75.8222	79.3758	
	FPFS-EC	94.2085	70.1592	67.0879	96.3468	74.7286	79.2559	
	FPFS-CMC	94.0483	69.1430	66.2550	96.2559	73.5161	78.6602	
	IFPIFSC	92.2406	63.4527	60.4435	95.1167	68.6435	72.5285	
	IFPIFS-HC	92.6453	64.7535	62.4497	95.4705	69.8072	73.6901	
	AIFPIFSC1	94.6825	76.8664	71.1918	96.6138	78.6652	81.3745	
	AIFPIFSC2	94.4892	76.1900	70.1926	96.4886	77.8141	80.6892	
Leaf	Fuzzy kNN	96.2016	32.5347	32.0444	98.0528	62.0025	32.7941	
	FPFS-kNN	97.8353	73.3337	67.4556	98.8794	73.8272	67.5294	
	FPFS-AC	97.8569	71.8471	67.6889	98.8910	74.4906	67.8529	
	FPFS-EC	97.8039	71.8566	66.9333	98.8633	73.7364	67.0588	
	FPFS-CMC	97.7686	71.2570	66.5444	98.8451	73.4777	66.5294	
	IFPIFSC	97.7608	71.0219	65.9667	98.8419	72.6249	66.4118	
	IFPIFS-HC	97.7176	70.1101	65.4611	98.8193	72.3277	65.7647	
	AIFPIFSC1	98.0784	75.5176	70.7778	99.0055	75.9554	71.1765	
	AIFPIFSC2	98.0843	75.2774	70.9778	99.0086	76.2688	71.2647	
Libras	Fuzzy kNN	95.8963	74.0815	69.2600	97.8017	70.1990	69.2222	
	FPFS-kNN	96.7037	79.3345	75.2267	98.2343	75.3681	75.2778	
	FPFS-AC	97.2815	82.1165	79.6200	98.5435	79.3057	79.6111	
	FPFS-EC	97.0667	80.8036	78.0400	98.4288	78.0580	78.0000	
	FPFS-CMC	96.9481	79.8471	77.1400	98.3655	77.2585	77.1111	
	IFPIFSC	95.9704	73.5069	69.9000	97.8417	70.0397	69.7778	
	IFPIFS-HC	96.1889	74.5579	71.4867	97.9589	71.0616	71.4167	
	AIFPIFSC1	97.3444	82.8129	80.1467	98.5777	79.7321	80.0833	
	AIFPIFSC2	97.2593	82.6086	79.5267	98.5321	79.3585	79.4444	
Mice	Fuzzy kNN	99.8608	99.4865	99.4429	99.9200	99.4470	99.4431	
	FPFS-kNN	100	100	100	100	100	100	
	FPFS-AC	100	100	100	100	100	100	
	FPFS-EC	100	100	100	100	100	100	
	FPFS-CMC	100	100	100	100	100	100	
	IFPIFSC	100	100	100	100	100	100	
	IFPIFS-HC	100	100	100	100	100	100	
	AIFPIFSC1	100	100	100	100	100	100	
	AIFPIFSC2	100	100	100	100	100	100	
HCV Data	Fuzzy kNN	97.0934	55.1835	48.1424	96.0803	64.1767	92.7334	
	FPFS-kNN	96.9913	56.7283	38.1858	90.3967	84.7871	92.4782	
	FPFS-AC	97.7862	70.9108	53.5449	94.7062	74.2838	94.4654	
	FPFS-EC	97.0867	59.1956	46.0910	91.5718	77.9663	92.7168	
	FPFS-CMC	97.0798	62.4317	47.7072	91.6225	75.4864	92.6994	
	IFPIFSC	97.7930	67.9679	55.7708	95.5895	75.1353	94.4824	
	IFPIFS-HC	97.8271	67.3802	54.1163	95.8067	76.3137	94.5677	
	AIFPIFSC1	98.1460	77.3406	62.4411	95.6001	77.2945	95.3649	
	AIFPIFSC2	98.0916	77.7644	62.1039	95.1745	76.7595	95.2289	
Column3c	Fuzzy kNN	80.4086	75.9644	61.3111	82.1301	63.4923	70.6129	
	FPFS-kNN	84.0430	73.2282	71.4111	87.7623	71.6751	76.0645	
	FPFS-AC	82.1290	69.5714	68.6000	86.5810	68.5903	73.1935	
	FPFS-EC	82.1720	69.6658	68.6444	86.6150	68.6118	73.2581	
	FPFS-CMC	81.8710	69.1010	68.1556	86.3683	68.1437	72.8065	
	IFPIFSC	85.0538	72.2589	71.4000	89.1168	71.3761	77.5806	
	IFPIFS-HC	84.3441	71.8038	70.3111	88.4744	70.4134	76.5161	
	AIFPIFSC1	85.7204	73.5563	72.8556	89.6705	72.7302	78.5806	
	AIFPIFSC2	85.6344	73.7498	72.4556	89.5333	72.3830	78.4516	
NewThyroid	Fuzzy kNN	95.0388	90.7974	87.8857	95.0752	87.9835	92.5581	
	FPFS-kNN	97.0853	96.7062	91.2889	95.7410	93.1413	95.6279	
	FPFS-AC	96.3721	93.8027	90.9937	95.3598	91.9502	94.5581	
	FPFS-EC	96.4961	93.7248	91.3365	95.5977	92.0784	94.7442	
	FPFS-CMC	96.6202	93.8859	91.6921	95.7696	92.4125	94.9302	
	IFPIFSC	97.6124	96.1328	94.4222	97.0730	94.9601	96.4186	
	IFPIFS-HC	97.8605	97.0014	94.8095	97.2816	95.5104	96.7907	
	AIFPIFSC1	98.0775	97.2697	95.1175	97.4475	95.8640	97.1163	
	AIFPIFSC2	97.9845	97.1825	94.8000	97.2937	95.6615	96.9767	
Vowel	Fuzzy kNN	99.2140	95.8955	95.6768	99.5677	95.6417	95.6768	
	FPFS-kNN	99.3352	96.5957	96.3434	99.6343	96.3159	96.3434	
	FPFS-AC	99.7337	98.6159	98.5354	99.8535	98.5312	98.5354	
	FPFS-EC	99.6474	98.1818	98.0606	99.8061	98.0533	98.0606	
	FPFS-CMC	99.6125	98.0247	97.8687	99.7869	97.8614	97.8687	
	IFPIFSC	98.9513	94.5523	94.2323	99.4232	94.2318	94.2323	
	IFPIFS-HC	99.2507	96.1076	95.8788	99.5879	95.8730	95.8788	
	AIFPIFSC1	99.7906	98.9147	98.8485	99.8848	98.8459	98.8485	
	AIFPIFSC2	99.7723	98.8142	98.7475	99.8747	98.7437	98.7475	
Iris	Fuzzy kNN	97.3778	96.4257	96.0667	98.0333	96.0430	96.0667	
	FPFS-kNN	97.5111	96.5657	96.2667	98.1333	96.2529	96.2667	
	FPFS-AC	97.4222	96.4202	96.1333	98.0667	96.1199	96.1333	
	FPFS-EC	97.4222	96.4202	96.1333	98.0667	96.1199	96.1333	
	FPFS-CMC	97.3778	96.3697	96.0667	98.0333	96.0521	96.0667	
	IFPIFSC	91.5111	88.0210	87.2667	93.6333	87.1189	87.2667	
	IFPIFS-HC	92.8000	90.1718	89.2000	94.6000	89.0407	89.2000	
	AIFPIFSC1	97.5111	96.5434	96.2667	98.1333	96.2543	96.2667	
	AIFPIFSC2	97.4222	96.4337	96.1333	98.0667	96.1199	96.1333	
Seeds	Fuzzy kNN	90.4444	87.2996	85.6667	92.8333	85.4920	85.6667	
	FPFS-kNN	92.8571	89.7479	89.2857	94.6429	89.1821	89.2857	
	FPFS-AC	93.7143	91.0210	90.5714	95.2857	90.4056	90.5714	
	FPFS-EC	93.1429	90.0824	89.7143	94.8571	89.5837	89.7143	
	FPFS-CMC	93.3016	90.3377	89.9524	94.9762	89.8107	89.9524	
	IFPIFSC	92.9524	89.7158	89.4286	94.7143	89.2777	89.4286	
	IFPIFS-HC	92.7302	89.5061	89.0952	94.5476	88.8379	89.0952	
	AIFPIFSC1	94.3175	91.7485	91.4762	95.7381	91.4237	91.4762	
	AIFPIFSC2	94.7619	92.3747	92.1429	96.0714	92.1077	92.1429	
Wireless	Fuzzy kNN	99.0725	98.1721	98.1450	99.3817	98.1485	98.1450	
	FPFS-kNN	95.2525	90.6801	90.5050	96.8350	90.5397	90.5050	
	FPFS-AC	95.5500	91.2345	91.1000	97.0333	91.0944	91.1000	
	FPFS-EC	94.6725	89.4593	89.3450	96.4483	89.3541	89.3450	
	FPFS-CMC	94.3875	88.8904	88.7750	96.2583	88.7865	88.7750	
	IFPIFSC	98.2200	96.4930	96.4400	98.8133	96.4425	96.4400	
	IFPIFS-HC	98.2675	96.6149	96.5350	98.8450	96.5404	96.5350	
	AIFPIFSC1	99.0100	98.0371	98.0200	99.3400	98.0204	98.0200	
	AIFPIFSC2	99.0875	98.1902	98.1750	99.3917	98.1746	98.1750	
Wine	Fuzzy kNN	82.4910	74.0454	72.4131	86.5646	72.5250	73.7365	
	FPFS-kNN	97.0063	95.6477	96.2381	97.8322	95.6542	95.5095	
	FPFS-AC	95.4011	93.8981	94.1831	96.6168	93.3926	93.1016	
	FPFS-EC	97.3090	96.1400	96.6000	98.0425	96.0995	95.9635	
	FPFS-CMC	97.1947	95.9917	96.4635	97.9589	95.9366	95.7921	
	IFPIFSC	98.0921	97.5585	97.6127	98.5773	97.3807	97.1381	
	IFPIFS-HC	98.3915	97.8120	97.9841	98.8081	97.7592	97.5873	
	AIFPIFSC1	98.1979	97.3359	97.7556	98.6979	97.4052	97.2968	
	AIFPIFSC2	98.5005	97.7649	98.1333	98.9283	97.8170	97.7508	
WholesaleR	Fuzzy kNN	64.5455	32.7706	32.1678	67.0562	36.7254	46.8182	
	FPFS-kNN	77.2273	34.0997	32.5751	66.0077	56.2602	65.8409	
	FPFS-AC	70.6061	33.4320	33.2949	66.6446	39.5025	55.9091	
	FPFS-EC	70.5909	33.1335	33.1258	66.3717	39.7357	55.8864	
	FPFS-CMC	70.5000	33.4180	33.2277	66.7968	39.1309	55.7500	
	IFPIFSC	70.5455	33.4457	33.4324	65.9316	38.5804	55.8182	
	IFPIFS-HC	70.1970	32.6805	32.7455	65.7756	39.0998	55.2955	
	AIFPIFSC1	71.0303	35.9047	35.8277	67.0121	38.4634	56.5455	
	AIFPIFSC2	71.0303	35.9170	35.9634	67.1598	38.1598	56.5455	
Breast Tissue	Fuzzy kNN	84.3146	55.5706	51.1056	90.6527	57.0971	52.9437	
	FPFS-kNN	88.8066	67.3580	65.0500	93.3213	69.8296	66.4199	
	FPFS-AC	90.1833	70.8637	69.6167	94.1207	72.5957	70.5498	
	FPFS-EC	88.6768	67.5787	64.8278	93.2355	71.9591	66.0303	
	FPFS-CMC	88.3968	66.5644	63.6722	93.0684	70.6789	65.1905	
	IFPIFSC	89.2323	69.9230	67.0278	93.5696	68.1063	67.6970	
	IFPIFS-HC	89.9942	71.4022	68.9444	94.0127	71.6861	69.9827	
	AIFPIFSC1	89.7792	71.5443	68.5667	93.8846	72.6934	69.3377	
	AIFPIFSC2	89.5613	71.2471	67.7611	93.7502	72.2527	68.6840	
Led7Digit	Fuzzy kNN	92.8360	65.5106	64.3707	96.0143	63.8773	64.1800	
	FPFS-kNN	92.3760	63.1691	62.1362	95.7535	63.2381	61.8800	
	FPFS-AC	92.8320	65.4730	64.4296	96.0116	64.0844	64.1600	
	FPFS-EC	92.8320	65.4730	64.4296	96.0116	64.0844	64.1600	
	FPFS-CMC	92.8280	65.4508	64.4010	96.0094	64.0602	64.1400	
	IFPIFSC	92.8320	65.4730	64.4296	96.0116	64.0844	64.1600	
	IFPIFS-HC	92.8640	65.5359	64.5760	96.0297	64.1747	64.3200	
	AIFPIFSC1	92.8480	65.3969	64.5455	96.0238	63.9295	64.2400	
	AIFPIFSC2	92.9000	65.6258	64.8141	96.0524	64.1907	64.5000	
Mean	Fuzzy kNN	91.1318	72.4892	70.2654	92.9899	74.6155	75.9551	
	FPFS-kNN	93.8310	79.0775	75.8214	93.9765	82.0681	83.3420	
	FPFS-AC	93.4057	80.1321	77.8002	94.2688	80.6779	83.2745	
	FPFS-EC	93.2752	78.7916	76.6913	94.0175	80.6779	82.6885	
	FPFS-CMC	93.1957	78.7142	76.5281	94.0077	80.1741	82.4181	
	IFPIFSC	93.2512	78.6349	76.5182	94.2836	79.2002	81.9587	
	IFPIFS-HC	93.4052	79.0292	76.9062	94.4012	79.8964	82.4427	
	AIFPIFSC1	94.3023	82.5859	80.2558	95.0420	82.4851	85.0485	
	AIFPIFSC2	94.3053	82.6093	80.1285	95.0217	82.3874	84.9823	
Note:

The best performance results are shown in bold.

Table 2 demonstrates that AIFPIFSC1 and AIFPIFSC2 precisely categorize the dataset “Mice Protein Expression” in the same manner as FPFS-kNN, FPFS-AC, FPFS-EC, FPFS-CMC, IFPIFSC, and IFPIFS-HC. Furthermore, according to all performance measures, AIFPIFSC1’s results for the datasets “NewThyroid”, “Vowel”, “Iris”, “Seeds”, “Wireless”, and “Wine” exceed the performance rates of 95%, 98%, 96%, 91%, 98%, and 97%, respectively. AIFPIFSC2’s results for the datasets “NewThyroid”, “Vowel”, “Iris”, “Seeds”, “Wireless”, and “Wine” exceed the performance rates of 94%, 98%, 96%, 92%, 98%, and 98%. Moreover, AIFPIFSC1 achieves the highest scores across all performance indicators in “Ecoli”, “Libras”, “HCV Data” (excluding Pre value), “Column3C” (excluding Pre value), “NewThyroid”, “Vowel”, and “Iris” (excluding Pre value). AIFPIFSC2 achieves the highest scores across all performance metrics in “Leaf” (excluding Pre value), “Seeds”, “Wireless”, “Wine” (excluding Pre value), “WholesaleR” (excluding Acc, MacF, and MicF value), and “Led7Digit”. As a result, the average performance outcomes presented in Table 2 suggest that AIFPIFSC1 and AIFPIFSC2 are more effective classifiers than other classifiers for the datasets under consideration.

Statistical analysis

This subsection conducts the Friedman test (Friedman, 1940), a non-parametric method, along with the Nemenyi test (Nemenyi, 1963), a post-hoc test, following the approach suggested by Demsar (2006). The process aims to evaluate all performance outcomes in terms of Acc, Pre, Rec, Spe, MacF, and MicF. The Friedman test creates a ranking based on performance for the classifiers across each dataset. Therefore, a rank of 1 indicates the top-performing classifier, followed by a rank of 2 for the next best, and so on. If classifiers exhibit identical performance, the test assigns them the mean of their potential ranks. Subsequently, it assesses the average ranks and computes χF2, which follows a distribution with k−1 degrees of freedom, where k represents the total number of classifiers. A post hoc analysis, such as the Nemenyi test, is applied to identify significant differences among the classifiers. Differences between any pair of classifiers that exceed the critical distance are regarded as statistically significant.

In the statistical analysis, the average ranking for each classifier is computed using the Friedman test. Nine classifiers, which means k=9, including AIFPIFSC1 and AIFPIFSC2, are compared concerning 15 datasets (denoted as N=15) for each of the six performance criteria. The Friedman test yields the following statistics for Acc, Pre, Rec, Spe, MacF, and MicF: χF2=59.27, χF2=48.50, χF2=62.13, χF2=46.78, χF2=52.15, and χF2=57.99, respectively.

With k=9 and N=15, the Friedman test indicates a critical value of 15.51 at a significance level of α=0.05 (for additional details, refer to Zar (2010)). As the Friedman test statistics for Acc (59.27), Pre (48.50), Rec (62.13), Spe (46.78), MacF (52.15), and MicF (57.99) surpass the critical value of 15.51, it implies a notable distinction in the performances of the classifiers. Consequently, the null hypothesis stating “There are no performance differences between the classifiers” is dismissed, allowing the application of the Nemenyi test. For k=9, N=15, and α=0.05, the critical distance is computed as 3.1017 following the Nemenyi test. Figure 1 illustrates the crucial diagrams generated by the Nemenyi test for each of the six performance metrics.

Figure 1 The essential diagrams resulting from the Friedman and Nemenyi tests at a significance level of 0.05 for the six performance criteria in the context of the classifiers mentioned above.

Figure 1 manifests that the performance differences between the average rankings of AIFPIFSC1 and those of IFPIFS-HC, FPFS-kNN, Fuzzy kNN, FPFS-CMC, IFPIFSC, and FPFS-EC, are more significant than the critical distance (3.1017). Besides, the performance differences between the average rankings of AIFPIFSC2 and those of FPFS-kNN, Fuzzy kNN, FPFS-CMC, IFPIFSC, and FPFS-EC are more significant than the critical distance (3.1017). Figure 1 shows that although the difference between the mean rankings of AIFPIFSC1 and FPFS-AC, as well as AIFPIFSC2 and FPFS-AC and IFPIFS-HC is less than the critical distance (4.0798), AIFPIFSC1 and AIFPIFSC2 outperforms them in terms of all performance measures. Therefore, AIFPIFSC1 and AIFPIFSC2 outperform them in all performance metrics.

Conclusion and future studies

This study introduced two adaptive machine learning algorithms that concurrently utilize seven pseudo-similarities over ifpifs-matrices, applying them to a data classification problem. Besides, it employed two soft decision-making methods, constructed by ifpifs-matrices to improve the aforesaid machine learning approaches. In this study, the input values λ1 and λ2, previously provided externally in the algorithm IFPIFSC, are determined adaptively based on the number of classes and instances. Implementing the ifpifs-matrices in the proposed algorithms resulted in a noticeable enhancement in classification Acc. The adaptive λ values played a crucial role in fine-tuning the classification process, relying on the specificities of each dataset. The adaptive nature of λ values in AIFPIFSC1 and AIFPIFSC2 proved pivotal in managing the intricacies and variabilities inherent in real-world data. This adaptability ensured the suggested algorithms remained robust and effective across diverse datasets and scenarios. The proposed algorithms are compared with well-known and state-of-the-art fuzzy/soft-set-based classifiers such as Fuzzy kNN, FPFS-kNN, FPFS-AC, FPFS-EC, IFPIFSC, and IFPIFS-HC. The performance results were utilized for a fair comparison and were statistically analyzed. Therefore, the current investigation demonstrates that the suggested approaches yield superior performance outcomes, making it a highly convenient method in supervised learning.

By dynamically adjusting lambda values and integrating ifpifs-matrices into two classification algorithms, this study contributes to the theoretical aspects of soft decision-making and machine learning. It provides a practical framework that can be employed in various real-world applications. The adaptability and efficiency of the proposed methods make it a valuable addition to machine learning, especially in scenarios where data uncertainty and dynamism are predominant.

Although our proposed algorithms can classify problems with intuitionistic fuzzy values, no fuzzy or intuitionistic fuzzy uncertain data exists in standard machine learning databases. Therefore, to show that the mentioned algorithms can successfully work on data with intuitionistic fuzzy uncertainties, we subject the data in common databases to fuzzification and intuitionistic fuzzification processes, and then apply the proposed approaches. One of the most well-known examples of intuitionistic fuzzy uncertainties can be expressed to explain the usefulness of the methods: If a detector x emits ten signals per second and produces six positive and four negative signals, this situation can be represented by the fuzzy value μ(x)=0.6. Since intuitionistic fuzzy sets extend fuzzy sets, they can also model this scenario using the intuitionistic fuzzy values μ(x)=0.6 and ν(x)=0.4. However, suppose the detector records six positive, three negative, and one corrupted signal. In that case, this cannot be described using fuzzy values alone but can be represented using intuitionistic fuzzy values as μ(x)=0.6 and ν(x)=0.3. These examples illustrate the superiority of intuitionistic fuzzy sets over traditional fuzzy sets. Furthermore, soft sets are required to address the challenge of determining the optimal location for constructing a wind turbine when processing data from detectors at various locations. As seen in this example, problems with intuitionistic fuzzy uncertainties are among the types of uncertainties we can encounter daily. Classical machine learning methods cannot work on data with such uncertainty, but the proposed machine learning approaches can efficiently work on issues such as those presented in the study. Moreover, there is no data with such uncertainty in known databases. Therefore, to compare the classical methods with our proposed approaches, we can obtain the performance results of classical machine learning methods and our proposed methods by converting these data into fuzzy and intuitionistic fuzzy values using the same classical data. The abilities and advantages of the proposed approach compared to the others can be summarized in Table 3.

Table 3 Comparison of the modeling ability and utilized concepts of the considered methods.

Classifer	Ref.	Crisp value	Fuzzy value	Int fuzzy value	Cla metric or sim	fpfs-p metric or sim	ifpifs-p metric or sim	fpfs DM	ifpifs DM	ADA	
Fuzzy-kNN	Keller, Gray & Givens (1985)	✓	✗	✗	✓	✗	✗	✗	✗	✗	
FPFS-kNN	Memiş, Enginoğlu & Erkan (2022b)	✓	✓	✗	✓	✓	✗	✗	✗	✗	
FPFS-AC	Memiş, Enginoğlu & Erkan (2022c)	✓	✓	✗	✓	✓	✗	✓	✗	✗	
FPFS-EC	Memiş, Enginoğlu & Erkan (2021)	✓	✓	✗	✓	✓	✗	✗	✗	✗	
FPFS-CMC	Memiş, Enginoğlu & Erkan (2022a)	✓	✓	✗	✓	✓	✗	✓	✗	✗	
IFPIFSC	Memiş et al. (2023)	✓	✓	✓	✓	✓	✓	✓	✗	✗	
IFPIFS-HC	Memiş et al. (2021)	✓	✓	✓	✓	✓	✓	✓	✗	✗	
AIFPIFSC1	Proposed	✓	✓	✓	✓	✓	✓	✓	✓	✓	
AIFPIFSC2	Proposed	✓	✓	✓	✓	✓	✓	✓	✓	✓	
Note:

Int, Cla, Sim, fpfs-p, ifpifs-p, DM, and ADA represent intuitionistic, classical, similarity, fpfs-pseudo, ifpifs-pseudo, decision making, and adaptive, respectively.

While the proposed algorithms demonstrate significant improvements in classification Acc by effectively incorporating ifpifs-matrices and adaptive λ values, an aspect warranting further attention is their time complexity. Comparative analysis reveals that our robust and adaptable algorithms perform slower than some benchmarked classifiers. This observation underscores the need for optimization in computational efficiency. Future research could focus on refining the algorithmic structure to enhance processing speed without compromising the Acc gains observed. Potential avenues might include integrating more efficient data structures, algorithmic simplification, or parallel processing techniques. Addressing this time complexity issue is crucial for practical applications, especially in real-time data processing scenarios. Further exploration of the scalability of the proposed methods for handling larger datasets could also be a valuable direction for subsequent studies. Moreover, future works can extend this research by integrating other forms of fuzzy and soft sets, e.g., bipolar soft sets (Mahmood, 2020) and picture fuzzy parameterized picture fuzzy soft sets (Memiş, 2023a) into machine learning (Li, 2024), further enhancing the capabilities and applications of adaptive machine learning methods based on soft decision-making.

Additional Information and Declarations

Competing Interests

The authors declare that they have no competing interests.

Author Contributions

Samet Memiş conceived and designed the experiments, performed the experiments, analyzed the data, performed the computation work, authored or reviewed drafts of the article, and approved the final draft.

Ferhan Şola Erduran analyzed the data, prepared figures and/or tables, authored or reviewed drafts of the article, and approved the final draft.

Hivda Aydoğan conceived and designed the experiments, performed the experiments, performed the computation work, prepared figures and/or tables, and approved the final draft.

Data Availability

The following information was supplied regarding data availability:

The Ecoli dataset is available at at UCI Machine Learning Repository: Nakai, K. (1996). Ecoli [Dataset]. UCI Machine Learning Repository. https://doi.org/10.24432/C5388M.

The Leaf dataset is available at UCI Machine Learning Repository: Silva, P. & Maral, A. (2013). Leaf [Dataset]. UCI Machine Learning Repository. https://doi.org/10.24432/C53C78.

The Libras is available at UCI Machine Learning Repository: Dias, D., Peres, S., & Bscaro, H. (2009). Libras Movement [Dataset]. UCI Machine Learning Repository. https://doi.org/10.24432/C5GC82.

The Mice Protein Expression dataset is available at UCI Machine Learning Repository: Higuera, C., Gardiner, K., & Cios, K. (2015). Mice Protein Expression [Dataset]. UCI Machine Learning Repository. https://doi.org/10.24432/C50S3Z.

The HCV Data is available at UCI Machine Learning Repository: Lichtinghagen, R., Klawonn, F., & Hoffmann, G. (2020). HCV data [Dataset]. UCI Machine Learning Repository. https://doi.org/10.24432/C5D612.

The Vertebral Column dataset is available at UCI Machine Learning Repository: Barreto, G. & Neto, A. (2005). Vertebral Column [Dataset]. UCI Machine Learning Repository. https://doi.org/10.24432/C5K89B.

The NewThyroid dataset is available at UCI Machine Learning Repository: Quinlan, R. (1986). Thyroid Disease [Dataset]. UCI Machine Learning Repository. https://doi.org/10.24432/C5D010.

The Connectionist Bench dataset is available at UCI Machine Learning Repository: Deterding, D., Niranjan, M., & Robinson, T. (1988). Connectionist Bench (Vowel Recognition-Deterding Data) [Dataset]. UCI Machine Learning Repository. https://doi.org/10.24432/C58P4S.

The Iris dataset is available at UCI Machine Learning Repository: Fisher, R. (1936). Iris [Dataset]. UCI Machine Learning Repository. https://doi.org/10.24432/C56C76.

The Seeds dataset is available at UCI Machine Learning Repository: https://doi.org/10.24432/C5H30K.

Charytanowicz, M., Niewczas, J., Kulczycki, P., Kowalski, P., & Lukasik, S. (2010). Seeds [Dataset]. UCI Machine Learning Repository. https://doi.org/10.24432/C5H30K.

The Wireless dataset is available at UCI Machine Learning Repository: Bhatt, R. (2017). Wireless Indoor Localization [Dataset]. UCI Machine Learning Repository. https://doi.org/10.24432/C51880.

The Wine dataset is available at UCI Machine Learning Repository: Aeberhard, S. & Forina, M. (1992). Wine [Dataset]. UCI Machine Learning Repository. https://doi.org/10.24432/C5PC7J.

The Wholesale customers dataset is available at UCI Machine Learning Repository: Cardoso, M. (2013). Wholesale customers [Dataset]. UCI Machine Learning Repository. https://doi.org/10.24432/C5030X.

The Breast Tissue dataset is available at UCI Machine Learning Repository: S, J. & Jossinet, J. (1996). Breast Tissue [Dataset]. UCI Machine Learning Repository. https://doi.org/10.24432/C5P31H.

The LED Display Domain dataset is available at UCI Machine Learning Repository: Breiman, L., Friedman, J., Olshen, R., & Stone, C. (1984). LED Display Domain [Dataset]. UCI Machine Learning Repository. https://doi.org/10.24432/C5FG61.

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
