# Peer review of "Adaptive machine learning approaches utilizing soft decision-making via intuitionistic fuzzy parameterized intuitionistic fuzzy soft matrices"

_PeerJ Computer Science, doi:10.7717/peerj-cs.2703_

## Round 0.1 · original submission · Major Revisions

Dear authors,

Reviewers have now commented on your paper. You will see that they advise you to make major revisions to your manuscript. If you are prepared to undertake the work required, I would be pleased to reconsider my decision.

If you decide to revise the work, please submit a list of changes or a rebuttal against each point that is being raised when you submit the revised manuscript.

Best wishes,
D. Pamucar

Reviewer 2 ·

Basic reporting

The manuscript entitled “Two adaptive machine learning methods based on soft decision-making via intuitionistic fuzzy parameterized intuitionistic fuzzy soft matrices” is well written, well presented and is correct mathematically. The manuscript can be considered for the next step subject to the following major revision.

Experimental design

See detailed report.

Validity of the findings

See detailed report

Additional comments

1. The title of the manuscript should be is very much ambiguous and it needs revision.
2. Abstract of the manuscript is too short to describe the proposed the study and its motivation. Further it has too many abbreviations, which makes it very less effective, hence it needs attention.
3. In introduction section authors should discuss
i. Literature review and historical background more effectively.
ii. Why they have chosen this topic.
iii. They should add a paragraph which show that the presented work is original.
iv. In this section there are many references which are discussed without any sequence, for example: [12, 13, 10] and [29, 25, 24]. Which is not geed and needs attention.
4. Algorithm 1 on page 3 is not stated well and it has some issues regarding its application, so this must be reconsidered.
5. The linguistic quality of the paper is not up to the standards. So please ask a native English speaker to help you to improve the linguistic quality of your paper.
6. What is the motivation to state a lot number of Definitions in one go, Definition 4 to Definition 19. Delete unnecessary definitions.
7. What is the source of Data in Table 2. Is it artificial?
8. The authors should do the comparison study to show that their work is more effective as compared to the existing work.
9. Name the “Conclusion” section as “Conclusion and Future Studies” and discuss the future studies of your work in bipolar fuzzy sets, bipolar soft sets, complex fuzzy sets and picture fuzzy sets.
10. To handle above consider the following, but not limited to, references:

A. Interval valued picture fuzzy Aczel–Alsina aggregation operators and their application by using the multiattribute decision making problem. Journal of Mathematics, 2023(1), 1707867.
B. Hamacher Prioritized Aggregation Operators Based on Complex Picture Fuzzy Sets and Their Applications in Decision-Making Problems, Journal of Innovative Research in Mathematical and Computational Sciences, 1(1), 33-54.
C. WASPAS Technique Utilized for Agricultural Robotics System based on Dombi Aggregation Operators under Bipolar Complex Fuzzy Soft Information, Journal of Innovative Research in Mathematical and Computational Sciences, 1(2), 67-95.
D. A Novel Approach towards Bipolar Soft Sets and Their Applications, Journal of Mathematics, Volume 2020, Article ID 4690808
E. Assessment of Solar Panel Using Multiattribute Decision‐Making Approach Based on Intuitionistic Fuzzy Aczel Alsina Heronian Mean Operator. International Journal of Intelligent Systems, 2023(1), 6268613.
F. Selection of Database Management System by Using Multi-Attribute Decision-Making Approach Based on Probability Complex Fuzzy Aggregation Operators. Journal of Innovative Research in Mathematical and Computational Sciences, 2(1), 1-16.

Reviewer 3 ·

Basic reporting

The work proposes two classification algorithms based on the use of intuitionistic soft fuzzy sets that have the potential to improve upon those already known. Initially, the comparison of the presented algorithms with the known ones is adequate. However, in my opinion, there are areas for improvement.
1. Motivation: The work should have a stronger motivation. In the first part, “The amount of data that has been made possible by technological advancement is always growing. This increasing amount of data may be analyzed and interpreted using machine learning, a technical improvement. Numerous industries frequently use this technology, including defense, finance, psychology, medicine, meteorology, astronomy, and space sciences (...). Several mathematical tools, such as fuzzy sets [35] and intuitionistic fuzzy sets [2], have been propounded to model these uncertainties,” the text should be more precise, indicating that the classification is supervised (assuming that in the data, the groups are already observed). It is also unclear what type of "uncertainty" the authors are referring to—whether it pertains to the data (not sharply observed but with approximate values) or to the modeling of the errors committed by the algorithms.
2. I understand that the classification method proposed by the authors is intuitionistic fuzzy soft. Since these methods are supposedly intended to be applied later by practitioners, it would be helpful to provide a practical explanation, with a simple example, illustrating what it means to classify using a matrix of intuitionistic fuzzy soft sets.
3. It is necessary to understand the information actually provided by the datasets. Beyond "#Instance" (I assume this refers to N), attributes, and classes, it would be helpful to summarize the real problem associated with each dataset in a single sentence. It is also needed giving indicating where these data can be obtained.
4. It is unclear how the authors handle membership and non-membership when calculating
5. Acronyms must be defined the first time they appear.
6. The writing must be improved as there are numerous typos. For example (this is just one example, as the paper is full of such issues), the hyphen between "if" and "set" is missing. Another example is in Definition 3, where the matrix m x n is misrepresented; the last column and last row are unnecessary. Note that these are just two examples from the beginning of the paper.
7. When citing papers, the numbers must be ordered. For example, on line 171, the sequence [15,33,14] should be [14,15,33].

Experimental design

It is adequate. However, the samples need to be explained in a phrase and the data need to be referenced.

Validity of the findings

Adequete

Additional comments

Be careful with typos.

---

## Round 0.2 · Minor Revisions

Dear authors,

Reviewers have now commented on your paper. You will see that they advise you to make minor revisions to your manuscript. If you are prepared to undertake the work required, I would be pleased to reconsider my decision.

If you decide to revise the work, please submit a list of changes or a rebuttal against each point that is being raised when you submit the revised manuscript.

Best wishes,
D. Pamucar

Reviewer 2 ·

Basic reporting

Authors have revised the manuscript comprehensively, carefully and authoritatively and all the points that I raised during the review process have been addressed, so now nothing is pending on my side. Hence I recommend that the manuscript should be accepted for publication in its current form.

Experimental design

N/A.

Validity of the findings

Excellent.

Additional comments

No comments.

Reviewer 3 ·

Basic reporting

The authors have addressed comments on the first draft. However, one question remains pending: The authors infer fuzzy and intuitionistic observations from crisp observations. In other words, they do not encounter fuzzy data but instead induce fuzziness in data that is crisp. You should clarify the usefulness of this approach.

Experimental design

See comment above

Validity of the findings

See comment above

---

## Round 0.3 · accepted · Accept

All the reviewers' comments have been addressed carefully and sufficiently, the revisions are rational from my point of view, I think the current version of the paper can be accepted.

Reviewer 3 ·

Basic reporting

No additional comments.

Experimental design

Correct

Validity of the findings

Adequate.